# Quantifying pupil-to-pupil SARS-CoV-2 transmission and the impact of lateral flow testing in English secondary schools

Trystan Leng [1,2✉], Edward M. Hill [1,2], Alex Holmes [1,3], Emma Southall[1,3], Robin N. Thompson [1,2], Michael J. Tildesley [1,2], Matt J. Keeling [1,2] & Louise Dyson [1,2]

A range of measures have been implemented to control within-school SARS-CoV-2 transmission in England, including the self-isolation of close contacts and twice weekly mass testing of secondary school pupils using lateral flow device tests (LFTs). Despite reducing transmission, isolating close contacts can lead to high levels of absences, negatively impacting pupils. To quantify pupil-to-pupil SARS-CoV-2 transmission and the impact of implemented control measures, we fit a stochastic individual-based model of secondary school infection to both swab testing data and secondary school absences data from England, and then simulate outbreaks from 31st August 2020 until 23rd May 2021. We find that the pupil-to-pupil reproduction number, $R_{school}$, has remained below 1 on average across the study period, and that twice weekly mass testing using LFTs has helped to control pupil-to-pupil transmission. We also explore the potential benefits of alternative containment strategies, finding that a strategy of repeat testing of close contacts rather than isolation, alongside mass testing, substantially reduces absences with only a marginal increase in pupil-to-pupil transmission.

[1] The Zeeman Institute for Systems Biology & Infectious Disease Epidemiology Research, School of Life Sciences and Mathematics Institute, University of Warwick, Coventry CV4 7AL, UK. [2] JUNIPER – Joint UNIversities Pandemic and Epidemiological Research, https://maths.org/juniper/. [3] Mathematics for Real-World Systems Centre for Doctoral Training, University of Warwick, Coventry CV4 7AL, UK. ✉email: trystan.leng@warwick.ac.uk

The ongoing Coronavirus-2019 (COVID-19) pandemic has seen unprecedented social restrictions placed upon populations globally. These have included general social distancing measures, the prohibition of households mixing socially, travel restrictions, the closure of pubs, restaurants, and non-essential shops, and have often involved school closures. However, given the importance of school attendance in future academic attainment, employment prospects, and income[1,2], school closures have been seen as a last resort[3]. Further, school closures may exacerbate educational inequalities[4,5], negatively impact children's mental health[6], and reduce access to much needed services for the most vulnerable children[7]. As countries emerge from a lockdown situation with the hope of relaxing social restrictions entirely, against a backdrop of increasing immunity in the population through vaccine uptake[8], key questions become how to minimise within-school transmission of SARS-CoV-2 whilst keeping schools open, and whether transmission can be mitigated using strategies that minimise the disruption caused by isolating close contacts of individuals who test positive.

SARS-CoV-2 infection rarely results in acute adverse health consequences for children[9–11]. While some children report long-lasting symptoms, such instances appear rare and to improve with time[12]. However, controlling transmission between children remains important because of potential onwards transmission to families, teachers and the wider community. Though previous studies tentatively suggest that transmission within schools does not drive transmission in the community[13,14], measures implemented at the school level may sometimes be sufficient to reduce the population-scale reproduction number ($R$) below 1[15]. Other studies have found that multiple within-school control measures in combination can mitigate within-school transmission[16,17] and the risk of onwards transmission from schools to the community caused by schools reopening[18].

In England, a range of school-level policies have been implemented to reduce within-school SARS-CoV-2 transmission[19]. In secondary schools, measures have included mask-wearing for pupils and teachers (mandated from 8th March-17th May 2021[20]), strict social distancing implemented through seating plans and the restriction of movement around schools, the implementation of 'bubbling' policies at the level of year groups or classes, and the temporary isolation of infected individuals and close contacts upon confirmation of a positive case.

Alongside these measures, since the reopening of secondary schools in England on 8th March 2021, both teachers and secondary school pupils have been strongly encouraged to participate in twice weekly mass testing using lateral flow device tests (LFTs). The aim has been to minimise the increase in transmission associated with keeping schools open by rapidly identifying asymptomatic and presymptomatic individuals. Participating pupils' first three tests prior to the 19th March 2021 were conducted in school; after this, tests were conducted at home. Any positive tests identified through home testing have been followed up by confirmatory polymerase chain reaction (PCR) tests, to minimise unnecessary absences from false positives. This policy has operated in tandem with a strategy of isolating close contacts of cases, to halt chains of transmission from infections that have already taken place.

While PCR tests must be processed in a laboratory, meaning results typically take up to 48 hours to return, LFTs can be taken at home and are capable of returning a result in 30 min. The rapidness of LFTs makes them ideal candidates for mass testing, and in the UK free LFTs have also been offered to the population at large since 9th April 2021. However, compared to PCR tests, LFTs are both less sensitive and less specific[21,22]. Despite their comparatively lower sensitivity, there is evidence emerging that such tests can play an important role in rapid testing; previous

studies have found that, when employing a mass testing strategy, the frequency of testing has a greater impact at reducing transmission than the sensitivity of the test[23]. Serial contact testing has also been suggested as a strategy to reduce within-school transmission that does not result in high levels of absences. Under serial contact testing, the close contacts of positively identified pupils are tested daily using LFTs for the next seven days, instead of isolating for ten days. Pilots within secondary schools in England to determine the efficacy of serial contact testing have been undertaken, which suggest that such a strategy is non-inferior to the isolation of close contacts in reducing within-school transmission[24]. A serial contact testing policy was considered as part of a national secondary school reopening strategy prior to the emergence of the Alpha (B.1.1.7) variant[25].

Previous studies have attempted to capture the impact of school-based measures on within-school transmission. In prior work, we considered the impact that strategies involving LFTs could have on both transmission and absences in secondary schools[26], finding that serial contact testing alone would be insufficient to control pupil-to-pupil transmission, but a policy of regular mass testing alongside serial contact testing could be more effective than the isolation of year group bubbles while reducing absences considerably. A parallel study by Kunzmann et al. considered the impact of such measures in primary schools, reaching a similar conclusion that serial contact testing alone would be insufficient to contain outbreaks, recommending a combination of mass testing and isolation of close contacts[27]. Other studies have explored the impact of alternative within-school control measures, such as PCR-based testing strategies[28], dividing classes into discrete cohorts[29,30], and mask-wearing[31]. Owing to the paucity of information surrounding SARS-CoV-2 transmission rates between children within schools, such studies typically consider a range of pupil-to-pupil transmission rates. Although these approaches are reasonable and valuable to compare the relative impacts of different school-based measures, quantifying the impact of implemented control measures on transmission requires knowledge of realistic levels of pupil-to-pupil transmission. Other studies have considered the impact of school closures or school-based control strategies on wider community transmission[15,32–34]. While these studies are important in understanding the contribution of schools to community transmission, such methods are insufficient to quantify the impact of measures on pupil-to-pupil transmission.

In this study, we extend our previously described stochastic individual-based model of SARS-CoV-2 transmission in a secondary school formed of exclusive year-group bubbles[26]. We incorporate realistic secondary school sizes and close contact group sizes (derived from Department for Education: Educational Setting Status data[35]), use S-gene negative data to incorporate the spread of the Alpha (B.1.1.7) variant and its impact on pupil-to-pupil transmission, and we use swab testing data from the wider population[36] to inform each school's probability of external infection from the local community and participation in mass testing, based on each school's lower tier local authority (LTLA). We fit this model using an Approximate Bayesian Computation (ABC) approach[37] to positive PCR and LFT time-series data from 11–16 year olds and the distribution of peak confirmed COVID-19 cases in secondary schools. From this fitted model, we estimate: the proportion of infections in secondary school pupils that occur due to pupil-to-pupil transmission; the pupil-to-pupil reproduction number ($R_{school}$); the impact of LFTs on incidence of infections at realised levels of participation; the benefit of higher participation levels of LFT mass testing; and the potential impact of serial contact testing instead of isolating close contacts. These analyses highlight approaches that can keep pupil-to-pupil transmission low while maintaining high levels of attendance, a

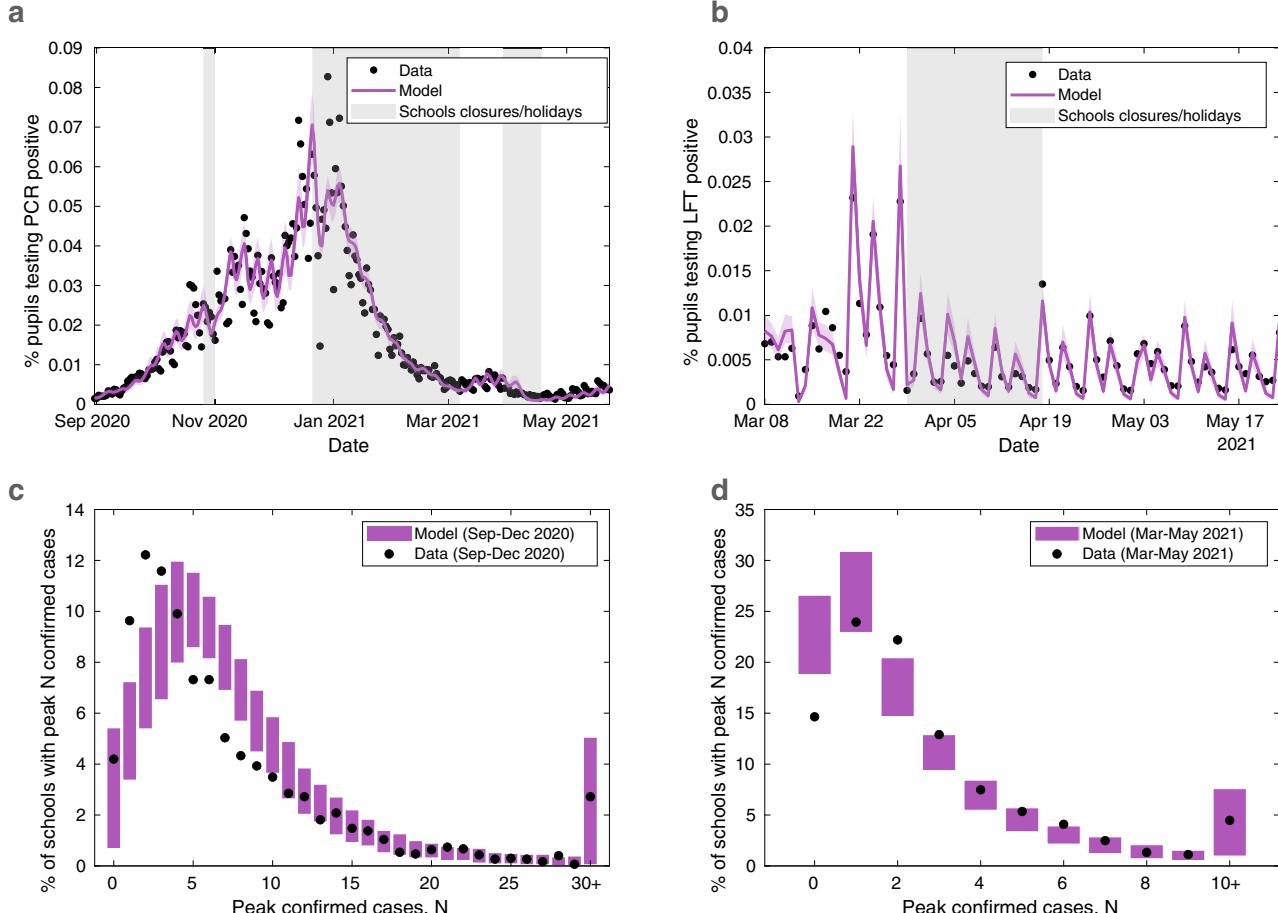

**Fig. 1 Fitting the model to testing and secondary school absences data.** The stochastic individual-based model is fitted to (**a**) the percentage of 11–16 year olds who test PCR positive (excluding confirmatory PCR tests) each day from 1st September 2020 to 23rd May 2021, (**b**) the percentage of 11–16 year olds who test LFT positive each day from 8th March 2021 to 23rd May 2021. Circles correspond to the data, with shaded intervals around mean model traces (solid lines) representing 95% prediction intervals in all plots. Shaded vertical grey regions represent time periods when schools were not fully open (either due to closures or school holidays). The model is also fitted to (**c**) the distribution of peak number of confirmed COVID-19 absences in secondary schools from 1st September 2020 to 18th December 2020, and (**d**) the distribution of peak number of confirmed COVID-19 absences in secondary schools from 8th March 2021 to 23rd May 2021. Circles denote the data and shaded blocks the 95% prediction interval estimated from the model. The plots above show the mean values obtained from 100 simulations in 2979 secondary schools, each with a distinct parameter set sampled from the posterior distribution.

difficult balance that is vitally important if we are to preserve the benefits of education during future waves of the pandemic.

## Results

**Model fit and parameter inference.** The fitted model matches well to the temporal nationwide data on PCR and LFT positivity in the school-aged cohort (Fig. 1a, b), whilst also providing a reasonable fit to the more challenging problem of matching the distribution of peak case numbers across schools in the September-December 2020 and March-May 2021 periods (Fig. 2c, d). We note that the model slightly underestimates the proportion of schools that had a low peak number of confirmed cases during the September-December 2020 period (Fig. 2c), while overestimating the proportion of schools with a low peak number of confirmed cases from March to May 2021 (Fig. 2d), despite being explicitly fitted to these data sources.

Two temporal trends are captured by our model: we infer an increase in pupil-to-pupil transmission due to the B.1.1.7 variant of approximately 81% (95% credible interval: 60–98%), and attribute a 44% (95% credible interval: 9–118%) increase in pupil-to-pupil transmission to falling adherence to within-school control measures after schools return from the October 2020

half-term break. The LFT data are best explained by a model that assumed only 38% (95% credible interval: 31–51%) of negative home LFTs are in fact reported. Imports of infection into the school from the community are estimated to be 59% (95% credible interval 15–90%) lower in rural areas compared to urban areas, allowing us to capture the spatial heterogeneity in reported infections (Supplementary Fig. 16). By modelling the isolation of close contacts using realistic group sizes that are isolated upon identification of a positive case, we also captured COVID-19 related absences through time (Supplementary Figs. 17 and 18).

**Quantifying pupil-to-pupil transmission.** We infer that the majority of pupil infections during term time and on school days occurred through contact with another pupil (Fig. 2a). The proportion of cases occurring between pupils increased through time, accounting for 45% (95% prediction interval: 26–63%) of all new infections in the September-October 2020 half term, 71% (95% prediction interval: 58–80%) of all new infections in the November-December 2020 half term, and 74% (95% prediction interval: 63–82%) of infections since schools reopened from 8th March 2021 until 23rd May 2021. The pupil-to-pupil reproduction number, $R_{school}$, mirrors the temporally increasing

**a** 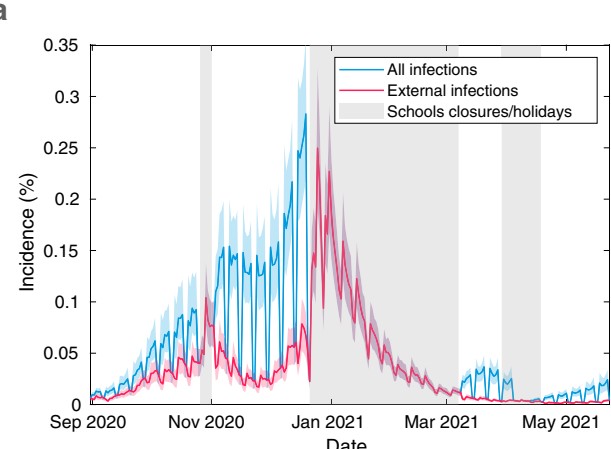

**b** 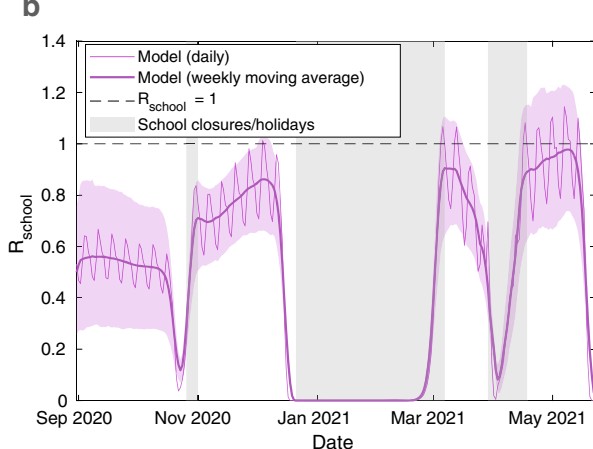

**Fig. 2 Incidence and $R_{school}$ from the fitted model.** We display time-series of (**a**) total incidence among pupils (blue) alongside incidence occurring through external (non-school) infections (red) (**b**) $R_{school}$ through time (thin line) alongside its seven-day moving average (thick line). Results obtained from 100 simulations in 2,979 secondary schools, each with a distinct parameter set sampled from the posterior distribution. In all panels, solid lines correspond to mean temporal profiles, shaded ribbons represent 95% prediction intervals and the shaded vertical grey regions represent time periods when schools were not fully reopen (either due to closures or school holidays).

trend–while $R_{school}$ remained well below one throughout the school term in 2020, it rose to an average daily value of 0.94 (95% prediction interval: 0.74–1.10) between 19th April and 9th May 2021, owing to the dominance of the more transmissible Alpha variant (Fig. 2b). Due to the stochastic nature of transmission and between-school variability, mean values of $R_{school}$ close to 1 are expected to generate a wide range of different sized localised within-school outbreaks.

**Impact of LFTs.** We compared our fitted model to a counterfactual scenario where mass testing was not introduced when schools reopened in March 2021. With an assumed LFT specificity of 99.97%, the data were best explained by a model that assumed only 38% of negative home LFTs are reported. With this level of underreporting, we estimated 36% (95% prediction interval: 29–42%) participation in LFT mass testing nationally from 8th March-23rd May 2021, though participation varied substantially between LTLAs (Supplementary Fig. 4b). Despite these relatively low levels of participation, mass testing has reduced incidence between pupils considerably (Fig. 3a, purple line), compared to a scenario where mass testing had not been introduced (Fig. 3a: blue line). We observe a reduction in $R_{school}$ attributable to mass testing, taking the average $R_{school}$ from 19th April-9th May 2021 from 1.09 (95% prediction interval: 0.90–1.26), which leads to exponential growth within schools, to 0.94 (95% prediction interval: 0.74–1.10) (Fig. 3b).

We also compare our fitted model to two other control scenarios. In the absence of additional measures targeted at close contacts, mass testing alone at realised levels of mass testing participation from March to May 2021 would be insufficient to reduce $R_{school}$ below 1 (Fig. 3b, red line), with an average daily $R_{school}$ from 19th April to 9th May 2021 of 1.03 (95% prediction interval: 0.85–1.18) but would still outperform the policy of isolation without mass testing. However, employing serial contact testing instead of the isolation of close contacts alongside mass testing (Fig. 3b, green line) only generates slightly higher values of $R_{school}$ over the period considered, with $R_{school}$ as a weekly moving average remaining below one for the majority of the interval.

Of the four control strategies considered, mass testing alongside the isolation of close contacts generates the largest number of pupil absences, peaking at 2.69% (95% prediction interval: 2.36–2.99%) in late March, in line with peak COVID-19 related

absences observed in data (Fig. 3c, purple line). This level of pupils being absent is attributable to a higher probability of detecting infection (due to mass testing) and the high level of absenteeism per case (due to isolation of close contacts). In contrast, a strategy of mass testing alone resulted in a peak of 0.19% (95% prediction interval: 0.16–0.22%) pupils being absent over the same period, while a strategy of mass testing alongside serial contact testing resulted in pupil absences peaking at 0.2% (95% prediction interval: 0.17–0.23%); these lower proportions a result of fewer absences per identified case.

**LFT participation counterfactuals.** Finally, for the three strategies involving twice weekly mass testing, we consider the necessary level of LFT participation to bring $R_{school}$ below 1. For this computation we assumed that all pupils scheduled to take a test on a given day do so with a probability $p$, and we varied $p$ from 0 to 1. As participation increases we find that $R_{school}$ falls linearly. Under a strategy of mass testing alongside isolation of close contacts, 18% participation (95% prediction interval: 0–50%) would have been required to bring the mean value of $R_{school}$ below one from 19th April to 9th May 2021 (Fig. 3d). The required participation increases to 24% (95% prediction interval: 0-57%) with a policy of mass testing with serial contact testing (assuming that all close contacts undergo serial contact testing). With a policy of twice weekly mass testing alone the required participation is higher still, with a mean estimate of 38% (95% prediction interval: 0–69%).

## Discussion

Epidemiological models matched to available data are vital tools to understand and predict the impacts of control measures. This study combines swab testing data collected from the wider population with secondary school absence data in England, recorded from August 2020 to May 2021, in a model of pupil-to-pupil SARS-CoV-2 transmission consistent with both data streams. We elucidate the impacts of the control measures that have been implemented (a combination of mass testing and isolation of school 'bubble' contacts), and we provide an insight into the potential impacts of alternative strategies.

Our results demonstrate that many cases in secondary-school aged children likely result from transmission from other

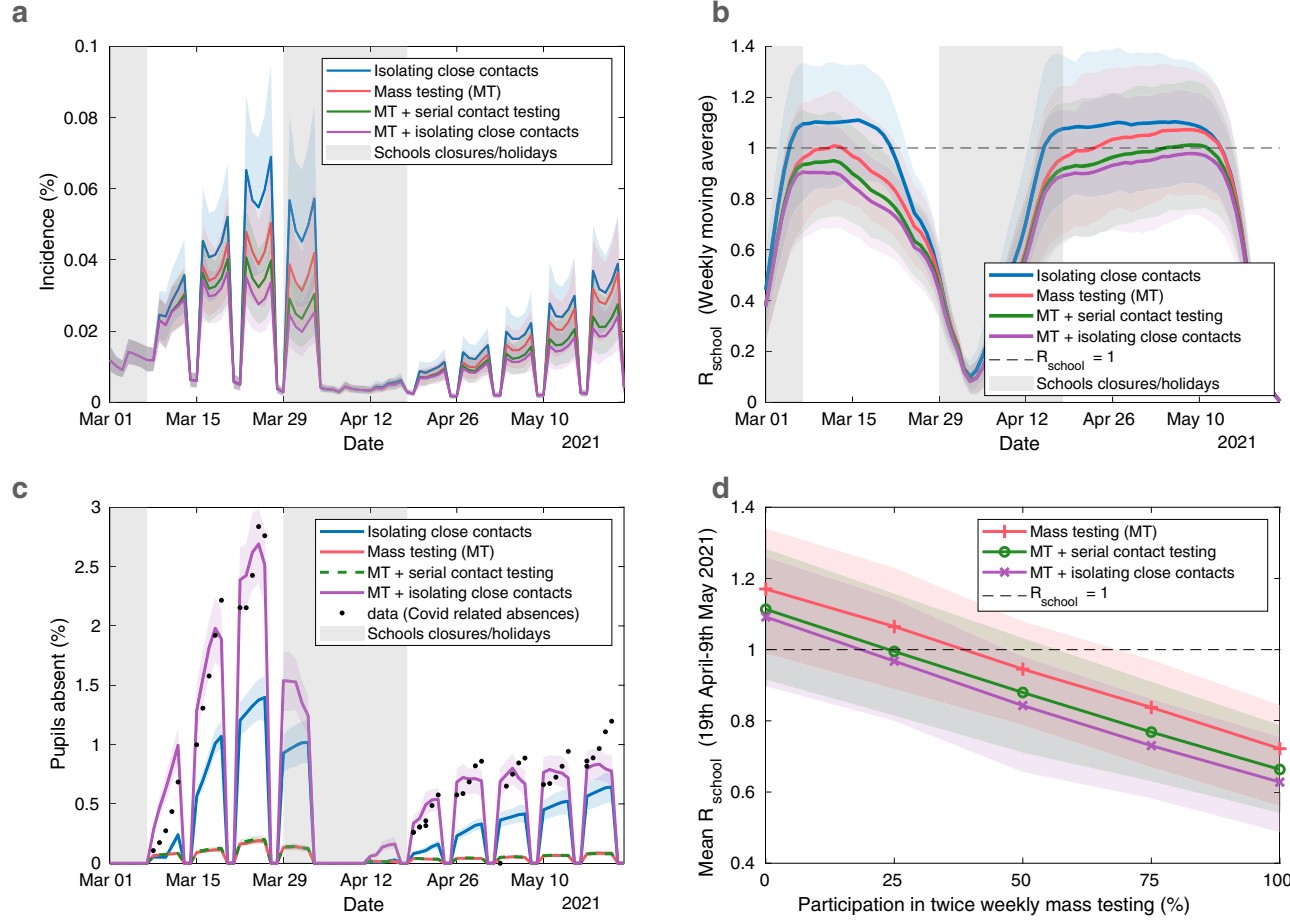

**Fig. 3 Quantifying the impact of LFTs on transmission and absences and the potential impact of alternative strategies.** Time-series under different intervention strategies of (**a**) incidence among pupils, (**b**) $R_{school}$ within secondary schools, and (**c**) the percentage of pupils absent; and (**d**) the average $R_{school}$ from 19th April 2021 to 9th May 2021 realised by different levels of participation in mass testing. We compare a policy of twice weekly mass testing and isolating close contacts (purple) to a strategy of isolating close contacts only (blue), twice weekly mass testing only (red), and twice weekly mass testing alongside serial contact testing (green). Results obtained from 100 simulations of 2979 secondary schools, each with a distinct parameter set sample from the posterior distribution. In all panels, solid lines correspond to the mean estimate, shaded intervals represent 95% prediction intervals and the shaded vertical grey regions represent time periods when schools were not fully reopen (either due to closures or school holidays). The data in panel (**c**) consists of the number of absences due to a confirmed case or a suspected case of COVID-19, and absences arising as a result of students told to isolate due to potential contact with a case of COVID-19 from inside their educational setting.

secondary school pupils, with such infections comprising approximately 45% of new infections in secondary-school aged children in the September-October 2020 half-term, 71% in the November-December 2020 half-term, and 74% from 8th March to 23rd May 2021. These results mirror the trends in community swab testing data collected from the wider population over the corresponding time periods, with a yet higher proportion of total positive PCR tests coming from secondary-school aged children in the three time periods (from Pillar 2 testing data, considering PCR tests excluding confirmatory tests, we calculate that 5.1% of positive tests were from 11-16 year olds from 31st August 2020 to 31st October 2020, 8.6% of positive tests were from 11–16 year olds from 1st November 2020 to 19th December 2021, and 9.2% of positive tests were from 11–16 year olds from 8th March 2021 to 23rd May 2021). This trend may be a consequence of the strict population control measures implemented from November-December 2020 and March-May 2021, coupled with schools remaining open over these periods[38]. The increase in $R_{school}$ is largely a consequence of the increased transmissibility of the Alpha variant, though our modelling suggests that reduced adherence to within-school control measures after the first half-

term of schools reopening may have also played a role. At the same time, our results suggest that transmission was not 'out of control' within secondary schools during the September-December 2020 term and from March to May 2021, as the estimated $R_{school}$ remained below one. Taken together, these results imply that sustained transmission between secondary school pupils has, over the time-period considered, required repeated external infection from the community, a result consistent with previous research indicating that within-school transmission has not driven community infection[13,14].

While we estimate there have been relatively low levels of LFT participation in secondary schools in England (approximately 36%, assuming a 99.97% LFT specificity), we demonstrate that LFTs have nevertheless played an important role in reducing incidence within secondary schools, which will have consequently reduced incidence in the wider community. Our results underline the importance of mass testing in controlling transmission within secondary schools, and highlights the potential benefits of even higher levels of participation. Our results therefore support the important role of mass testing via LFTs in reducing transmission[23,24,26], despite their lower sensitivity compared to PCR tests[21].

Our results also highlight that strategies involving isolating large numbers of close contacts lead to considerable levels of school absences[26]. In the context of minimising educational disruption from pupil absences, we considered the likely impact of alternative strategies. While mass testing alone (with approximately 36% participation) would have been insufficient to keep $R_{school}$ below one over the course of schools reopening from March to May 2021, a policy of regular mass testing alongside serial contact testing is predicted to be almost as effective at reducing pupil-to-pupil transmission as regular mass testing alongside isolating close contacts, but with considerably fewer absences.

When considering serial contact testing, this study optimistically assumed that all pupils agree to participate in daily testing if identified as a close contact of a positive case. In practice, some pupils may not participate. In a recent trial to determine the efficacy of serial contact testing in secondary schools, only 42.4% of identified contacts actively participated[24]. If non-participating close contacts must self-isolate then low compliance will increase pupil absence, whereas if non-participating close contacts can remain in school then the amount of pupil-to-pupil transmission would increase. The most suitable option depends both on expected levels of participation and the intended goals of such a policy, demonstrating that clear protocols and well-defined aims are paramount to the successful implementation of such targeted strategies[39].

Any mathematical modelling study is a simplification of the real-world, and necessarily involves assumptions. Accordingly, our study has several limitations. The following paragraphs discuss the study's limitations regarding: (i) transmission and contact structure, (ii) mass testing, (iii) vaccination, (iv) other aspects of school transmission, and (v) the interpretation of inferred parameters.

Although our model captures the impact of community prevalence on pupil-to-pupil transmission, it does not capture the impact of pupil-to-pupil transmission on community prevalence. In reality, within-school epidemics may increase community prevalence in extremely local areas (smaller than that of an LTLA), which would then be expected to increase transmission in schools as a damped feedback loop. Our study assumed three levels of mixing, with pupils transmitting infection at high rates to their close contacts, at a lower rate to other pupils in their year, and at a yet lower rate to pupils in other years. Assuming that schools implemented consistent isolation policies throughout each term, we also assumed that close contact group sizes were of a fixed size for each school. While this assumption allowed us to successfully match to absence data throughout both terms, we acknowledge it limits the heterogeneity in contact patterns within schools. In reality, transmission rates are likely to be heterogeneous within schools, both as a consequence of heterogeneous contact patterns and because transmission rates are likely a function of peak viral load[40], which varies between individuals. While previous studies undertaken prior to the COVID-19 pandemic have attempted to record contact mixing patterns within schools[41–43], the implementation of rigid social distancing measures within schools mean that such studies are not of direct use in the context of COVID-19. The CoMix study has surveyed social contacts in the UK during the COVID-19 pandemic and has been used to infer age-dependent mixing matrices[44], though these data are not directly informative of contact structure within schools. A deeper understanding of the interplay between contact network structure within schools and the success of control measures would be an important contribution going forward. Previous studies accounting for heterogeneity in transmissibility through the incorporation of within-host viral dynamics[27] obtained similar results to our previous study[26], though the inclusion of heterogeneity may impact the peak sizes of epidemics in schools[45].

We also assumed, due to data aggregation, that the proportion of secondary school pupils taking an LFT test on a given day is equivalent to the local proportion of 10–19 year olds in that school's LTLA taking an LFT on a given day, i.e. we assumed participation was homogeneous across schools within a region. In reality, there may be significant heterogeneity between schools even within a local area. Further, we assumed that all pupils have an equal probability of taking an LFT to satisfy a given level of participation; in reality, some pupils may consistently take LFTs, while others may not. Including such heterogeneities would be expected to increase the variability in cases between schools, while persistent non-participation of some pupils will increase $R_{school}$ as asymptomatic infections in such individuals will not be detected. As a further complication, underreporting of negative tests is an important but unknown factor which could vary both regionally and temporally–which we estimated as a single level of underreporting. These complications highlight the importance of accurate reporting of all test results, as accurate estimates of LFT usage are integral in understanding their impact.

Vaccination is not considered explicitly in the model. By 23rd May 2021, no COVID-19 vaccine had been approved in the UK for general use in those aged 12–17 years, but vaccines were available to adolescents within that age group who were classed as clinically vulnerable and/or who were 18 years old. In England, 0.02% of 12–15 year olds and 4.3% of 16–17 year olds had received at least one dose of vaccine by 23rd May 2021[46]. A higher proportion of 18 year olds had likely received one dose by this date, as 16.8% of 18–24 year olds had received at least one dose, though not all 18 year olds attend secondary school. Because vaccine uptake prior to 23rd May 2021 was relatively limited amongst secondary school pupils, we would not expect the action of vaccination to have had a substantive impact on our results. Since 13th September 2021, a vaccine has been approved for all 12–17 year olds[47], and consequently the inclusion of vaccination explicitly may be an important factor in future models of transmission within secondary schools. The impact of vaccination on external transmission is implicit in our model, as the probability of external transmission depends upon community infections, which in turn depends upon local vaccine uptake. However, we do not account for heterogeneities in vaccine uptake within an LTLA, which may impact a pupil's probability of external infection.

Results should be interpreted in reference to the underlying model structure. Our model focuses on transmission that occurs between secondary school aged pupils. While one might presume that infection occurs within the secondary school setting, this may not be the case–secondary school pupils will also mix outside of school. This point is particularly relevant in the context of closing schools, which would not be expected to reduce pupil-to-pupil transmission to zero, and may increase mixing between pupils in other settings. Teachers and other staff members are not included explicitly in our model. Accordingly, our model assumes the impact of transmission from teachers to pupils comprises some proportion of 'external' infections. Previous studies have suggested there is no evidence that teachers are at greater risk of infection or hospitalisation than other key workers[48,49], though their relative risk may be context dependent. Provided appropriate data were available to parameterise the interaction between teachers and pupils, further research exploring the impact of within-school epidemics on infections in teachers and staff, and conversely the impact of teachers and staff on within-school epidemics, would be valuable.

Similarly, inferred parameters should be interpreted cautiously and in the context of the underlying model assumptions. For

example, the model fits the proportion of infected secondary school pupils who are symptomatic and the relative infectiousness of asymptomatic individuals, under the assumption that all symptomatic individuals seek a test upon symptom onset. In reality, not all symptomatic individuals will seek a test, and may remain in school throughout their infectious period. Consequently, the 'true' proportion of pupils who are symptomatic will be higher than the value obtained through model fitting, while the 'true' relative infectiousness of asymptomatic individuals will be lower. Similarly, levels of underreporting are inferred under the assumption that LFTs during mass testing schools have had a 99.97% specificity (in line with contemporary estimates[50,51]). Lower assumed LFT specificity would yield lower levels of underreporting, as more false positives would occur for a lower number of tests. Model parameters may be impacted by the emergence of new variants not included in the model, such as the Delta (B.1.617.2) variant. More transmissible variants would be expected to increase pupil-to-pupil transmission rates, but may also impact the relative infectiousness of asymptomatic pupils and the proportion of pupils who display symptoms if the new variant changes the symptomatology of the virus (though no significant changes in symptomatology were observed between the wild-type and the Alpha variant[52]). Relatedly, the emergence of new variants may impact the generation time distribution of SARS-CoV-2[53], which informs the infectiousness profile of individuals within the model, while test sensitivity and specificity may also be impacted[54]. Such changes may impact the relative effectiveness of different control measures.

While our model considers a time period before the Delta variant dominated infections in the UK[55], and we do not explicitly consider the Delta variant in the model, we can nevertheless consider the implications of our work in the context of new, more transmissible variants. With $R_{school}$ approximately equal to one in mid-May 2021, more transmissible variants such as the Delta variant (estimated to be 60% more transmissible than the Alpha variant[56]) could increase $R_{school}$ substantially above one. If such an increase were to occur, within-school epidemics would occur more frequently. In turn, this may result in high levels of absences as a consequence of high numbers of cases amongst pupils. Further, our model considers a time-period when stringent within-school control measures were implemented. Prior studies have demonstrated that within-school measures can mitigate within-school transmission effectively[16], and high attack rates in schools have been observed when such measures have not been in place[57]. Any relaxation of within-school control measures would therefore likely result in further increases in pupil-to-pupil transmission and hence $R_{school}$. Because of these factors, far higher participation with lateral flow testing may be necessary to mitigate within-school infections in the future, especially to offset the impact of 60% extra transmission associated with the Delta variant. A range of socioeconomic factors are known to impact LFT participation, including the fear of loss of income that could result from a household required to self-isolate[22]. Policy makers should therefore consider practical strategies to increase uptake of LFTs across all sociodemographic groups, especially for any future strategy that does not include the isolation of close contacts[58]. Future studies considering the impact of Delta and future variants on transmission within secondary schools, and whether strategies utilising mass testing remain capable of mitigating transmission while keeping absence levels low, would be a valuable line of further research.

Our analyses have only considered the impact of LFTs in the context of secondary schools. In England, practical considerations have dictated that mass testing has not been extended to primary school children[58]. If the isolation of close contacts is halted in primary schools, this raises the question of which control measures to implement instead, whether they will be sufficient to control pupil-to-pupil transmission, and whether they are practical to implement for that age range.

Although we have demonstrated that twice weekly lateral flow testing has reduced pupil-to-pupil SARS-CoV-2 transmission in England since its introduction for secondary school pupils, keeping $R_{school}$ on average below one from March to May 2021, our results also indicate the fragility of the situation. With $R_{school}$ only just below one, increases in pupil-to-pupil transmission, either because of more transmissible variants or a relaxation of within-school distancing measures, are expected to result in substantial within-school outbreaks. We have shown the potential of serial contact testing alongside twice weekly mass testing to control pupil-to-pupil transmission while minimising the disruption caused by pupil absences, and the increased effectiveness of mass testing strategies at higher LFT uptake. Alternative strategies to isolating all close contacts are worth considering. Strategies involving the targeted use of LFTs may strike a balance between lowering transmission and reducing pupil absences.

## Methods

We extended our existing stochastic individual-based model of secondary schools formed of year-group bubbles[26], to quantify the level of pupil-to-pupil transmission of SARS-CoV-2 in secondary schools in England from 31st August 2020 to 23rd May 2021, and assessed the impact of LFTs on pupil-to-pupil transmission from 8th March 2021 to 23rd May 2021.

Absences data associated with COVID-19, aggregated at the level of schools, were provided through a data-sharing agreement between the Department for Education and the authors' institutions. The ethics of the use of these data for these purposes was agreed by the Department for Education with the UK Government's SPI-M(O)/SAGE committees. Anonymised community testing data, including details on the type of test and results, were provided through a data-sharing agreement between Public Health England and the authors' institutions. The ethics of the use of these data for these purposes was agreed by Public Health England (now UK Health Security Agency) with the UK Government's SPI-M(O)/SAGE committees.

Since September 2020, schools have recorded absences data associated with COVID-19[35]. These data contain, for each school, its location (defined as its LTLA), the number of students on roll, the number of confirmed COVID-19 cases known to each school through time and the number of pupils isolating due to close contacts with suspected COVID-19 cases within an educational setting.

We constructed a synthetic population of secondary schools matching recorded population sizes and simulated the spread of SARS-CoV-2 between pupils within a secondary school, coupled with a policy of isolating close contacts as implemented in England throughout the time frame. Our model focused on transmission between secondary school pupils on school days; teachers and other members of staff were not included explicitly in the school population. For each school, the level of reported infection in the associated LTLA acted to seed new infections within schools. Infections then spread stochastically to close contacts within the year group, other pupils within the year group and to other pupils within the school at three different rates (Fig. 4); these rates were increased to mimic the spread of the Alpha (B.1.1.7) variant through each LTLA from November 2020.

Within the model, case detection was conducted through two testing processes, mirroring testing policies implemented in secondary schools in England over the time-period considered. From 31st August 2020 onwards, symptomatic pupils sought a PCR test upon symptom onset. Those testing positive entered isolation for the next ten full days[59]. From 1st March 2021 onwards, alongside symptomatic pupils taking a PCR test upon symptom onset, positive cases were also detected through asymptomatic LFT mass testing. Pupils who tested positive using an LFT entered isolation, with the outcome of a confirmatory PCR test then determining whether the pupil remained in isolation (for a period including the day the LFT was taken and the next ten full days[59]). Prior to the introduction of asymptomatic LFT mass testing, only symptomatic pupils were detected. Secondary schools isolated the close contacts of infected pupils upon a pupil testing positive to a PCR test (either through self-seeking or as a confirmatory test to a positive LFT) for ten days following the day of last contact[60]. Close contact group sizes were defined as the number of pupils asked to self-isolate upon identification of a positive case. Schools were assumed to implement consistent isolation policies throughout each term, meaning that close contact group sizes were fixed within each school. Close contact group sizes were inferred from Department for Education: Education Setting Status data in order to capture the proportion of absences that occur through time (Supplementary Text 2). Both PCRs and LFTs were modelled with imperfect sensitivity that varies over an individual's period of infection[61]. We assumed 100% PCR specificity (in line with data indicating that false PCR positives are very rare[62]) and 99.97% LFT specificity (in line with estimates obtained from analysis of LFTs taken in secondary schools in England[51]).

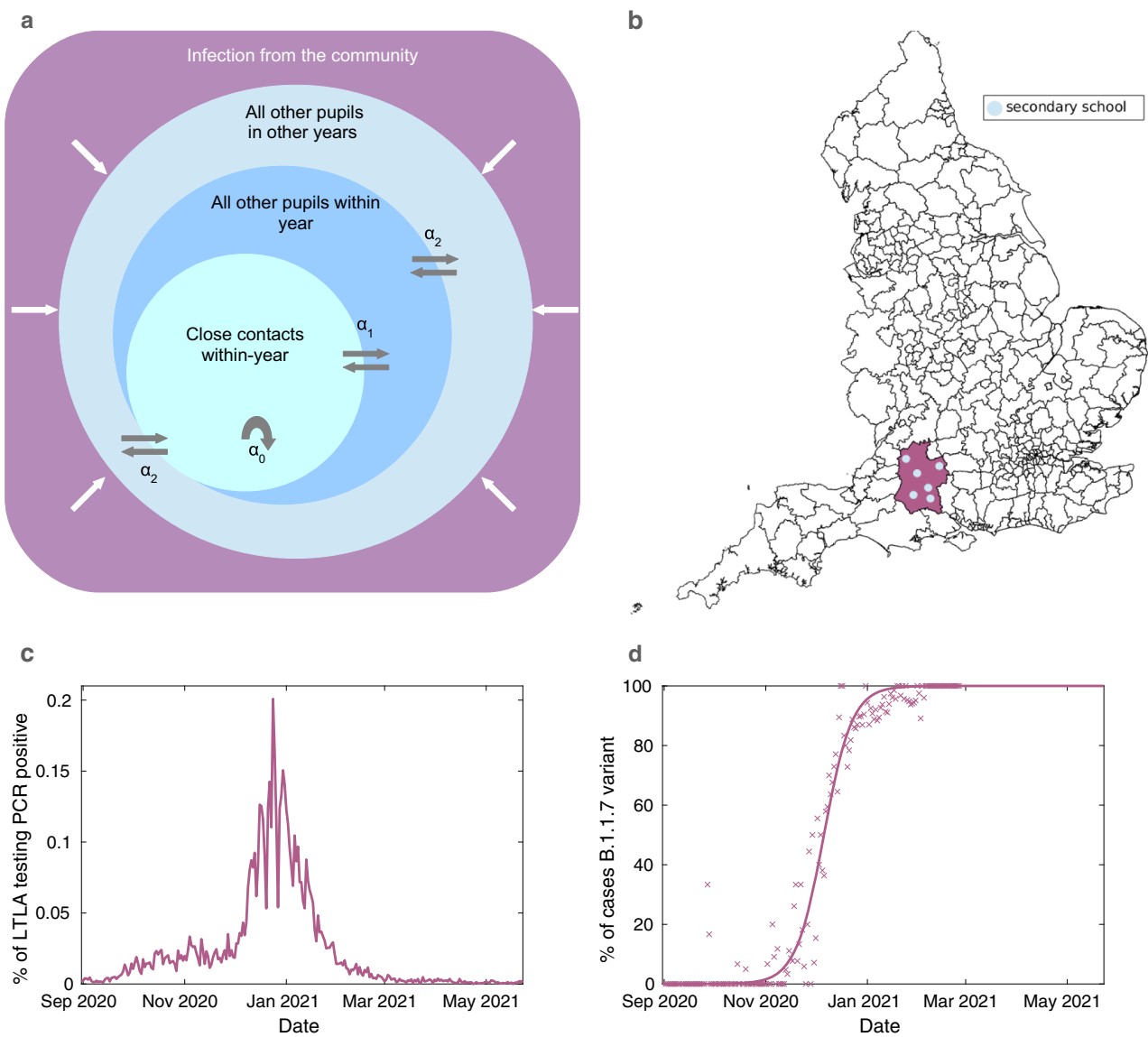

**Fig. 4 Overview of the individual-based model components. a** A schematic of the modelled within-school mixing structure. Within a school, pupils interact with close contacts in their year at baseline rate $\alpha_0 = 1$, with other pupils within their year at a relative rate $\alpha_1$, and interact with other pupils in other years at a relative rate $\alpha_2$, where $0 \leq \alpha_1, \alpha_2 \leq 1$. **b** England divided by LTLA, with an example LTLA highlighted in purple. Each school is situated within an LTLA, which determines its probability of infection from the community, its relative frequency of the B.1.1.7 (Alpha) variant, and LFT uptake. Each LTLA contains multiple secondary schools, shown as blue dots (the number of blue dots shown is illustrative rather than an accurate depiction of the number of secondary schools in the highlighted LTLA). **c** A time-series of the percentage of that LTLA's population testing positive on a PCR test on that day. A pupil's probability of external infection on day $t$ depends upon prevalence in the community, which we assume to be proportional to the proportion of the population in that LTLA testing PCR positive on day $t + 5$. **d** A time-series of the fitted estimate of the relative frequency of the B.1.1.7 variant in the example LTLA. The expected number of secondary infections from infected pupils depends upon the proportion of cases that are of the (more transmissible) B.1.1.7 variant, which varies through time and is dependent on the LTLA the school is situated within. Cross markers indicate the percentage of PCR tests from an LTLA that return an S-gene negative result out of those that return an S-gene status. Our model does not consider the impact of the B.1.617.2 (Delta) variant, which became the dominant variant in circulation during late May 2021, i.e. occurring beyond the time horizon of our analyses.

The model was fitted to swab testing data recorded in 11–16 year olds collected in the wider population from both PCR tests and LFTs, referred to as Pillar 2 data, which we took as a proxy for positive testing rates within secondary school pupils. The model also fitted school-level absences data recorded in the Department for Education: Educational Setting Status data. Another use of the Pillar 2 data was to inform the level of community infection and level of LFT participation within each LTLA in the model. The growth of the Alpha variant within an LTLA was inferred from the proportion of S-gene failures reported within an LTLA through time. We discuss the model structure, fitting procedure and data used in more detail in the Supplementary Material (Supplementary Text 1-5).

Our model fits 10 parameters in total (Supplementary Table 1): (1 and 2) Baseline pupil-to-pupil transmission rates for each school; (3) a scaling factor increasing pupil-to-pupil transmission after the October 2020 half-term; (4) the increased transmissibility of the Alpha variant; (5) a scaling constant determining the probability of external infection to a pupil given the level of community infection; (6) a scaling factor reducing the probability of external infection for schools in rural areas; (7) a scaling factor increasing the probability of external infection during school holidays; (8) the relative infectiousness of asymptomatic pupils; (9) the proportion of infected pupils who become symptomatic; and (10) the level of underreporting of negative home LFTs.

In the main analyses, we assumed pupils mainly mix with their year-group close contacts and assumed very low levels of mixing between years (Fig. 4a); we explored sensitivity to this assumption in Supplementary Text 8 and obtained qualitatively similar results (Supplementary Figs. 19–21).

Using this model, we compared four different strategies for controlling pupil-to-pupil transmission; these were simulated over the period 1st March 2021 (the week

prior to schools reopening after the 2021 lockdown) to 23rd May 2021 (when the Delta variant began to spread widely):

1. *Spring 2021 strategy*. The policy of mass lateral flow testing (used from 1st March to 23rd May 2021) and confirmatory PCR testing, with imperfect sensitivity, followed by the isolation of all close contacts of an infected case.
2. *Isolation of close contacts only*. A counterfactual scenario, with no mass testing. Schools continue to implement the isolation of close contacts policy used by schools from 31st August to 18th December 2020. Because there is no mass testing, only symptomatic pupils are detected.
3. *Mass testing only*. From 1st March 2021, the week prior to schools fully reopening, pupils undertake twice weekly mass testing (calibrated to obtain realistic levels of uptake). However, identification of a positive case leads to no further action, other than isolating the confirmed positive individual.
4. *Mass testing and serial contact testing*. Alongside mass testing, and upon identification of a positive case via a PCR test (either from a symptomatic pupil seeking a PCR test or as a confirmatory test from a positive LFT), that pupil's close contacts take LFTs for the next seven days following their last contact with the positive case (serial contact testing). It is assumed that all pupils participate in serial contact testing when identified as a close contact of a positive case.

To understand the impact of different strategies on pupil-to-pupil transmission and pupil absences, we used three main model outcome measures:

1. *Incidence*. Tracking the incidence of infections, and the impact of control measures on that incidence, is a natural measure to judge the benefit of control measures in reducing infections. However, incidence also depends on within-school prevalence and community prevalence, so does not (directly) inform whether pupil-to-pupil transmission is under control in secondary schools. Further, by tracking whether new infections occur through a contact with another pupil or an external contact, we can estimate the proportion of infections that occur between pupils during term time, and whether this has changed over the time-period considered.
2. *Pupil-to-pupil reproduction number* ($R_{school}$). A case reproduction number, defined as the number of secondary infections resulting from contact with an individual infected on date $d$ (over their entire courses of infection) divided by the number of individuals infected on date $d$. This outcome measure tells us whether pupil-to-pupil transmission is under control, indicated by $R_{school} < 1$, and how this has changed through time in the context of emerging variants and changing control measures.
3. *Percentage of pupils absent*. The percentage of all modelled pupils absent on a given day, either because they have tested positive or because they are a close contact of a positively identified individual in a school implementing an isolation of close contacts policy. This outcome measure is useful to understand the impact of different strategies on pupil absences (and therefore the potential disruption to pupil attendance).

**Reporting summary**. Further information on research design is available in the Nature Research Reporting Summary linked to this article.

## Data availability

Data from the Department for Education Educational Settings database were supplied after anonymisation under strict data protection protocols agreed between the University of Warwick and the Department for Education in the UK. The ethics of the use of these data for these purposes was agreed by the Department for Education with the Government's SPI-M(O)/SAGE committees. Due to the sensitive nature of the data, they can only be made available by DfE through a data-sharing agreement directly with the user, and so are not publicly available. For queries regarding access to COVID-19 related pupil absences data aggregated at the level of schools, contact Datarequests.COVID@education.gov.uk.

Public Health England (PHE) collected data in a centralised database, which included details on the type of test and results. PHE provided anonymised data to contributors of the Scientific Pandemic Influenza Group on Modelling (SPI-M) as part of the COVID-19 response under a data sharing agreement between PHE and the authors' institutions. The above data contain confidential information, with public data deposition non-permissible for socio-economic reasons. Due to the sensitive nature of the data, they can only be made available by PHE through a data-sharing agreement directly with the user. For queries regarding access to daily SARS-CoV-2 community testing data stratified by LTLA, type of test, and result of test (in one year age-bands for positive tests, and five year age-bands for negative tests), contact foundry.support@england.nhs.uk - each application for access to data will be considered on a case-by-case basis, and more information with regards to terms of access can be found at https://www.england.nhs.uk/ourwork/tsd/data-info/.

Positive case data in five year age-bands at the LTLA level are publicly available via the UK Coronavirus dashboard: https://coronavirus.data.gov.uk/details/cases, while absences data at the LTLA level are available from the Department for Education https://www.gov.uk/government/statistics/pupil-absence-in-schools-in-england-autumn-2020-and-spring-2021.

Population sizes stratified in each LTLA are available from the Office for National Statistics population estimates. In this study, we use the mid-year estimates from 2019, which are available at: https://www.ons.gov.uk/peoplepopulationandcommunity/populationandmigration/populationestimates/datasets/populationestimatesforukenglandandwalesscotlandandnorthernireland.

## Code availability

Code for the model and model fitting is available at: https://github.com/tsleng93/SchoolReopeningStrategies/tree/main/FittedModel.

Archived code at time of publication: https://doi.org/10.5281/zenodo.5898631.

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

## Acknowledgements

This work has been supported by the Engineering and Physical Sciences Research Council through the MathSys CDT [grant numbers EP/S022244/1 and EP/L015374/1]; and by the Medical Research Council through the COVID-19 Rapid Response Rolling Call [grant number MR/V009761/1]. T.L., M.J.K., L.D. and M.J.T. were supported by Medical Research Council through the JUNIPER modelling consortium [grant number MR/V038613/1]. The funders had no role in study design, data collection and analysis, decision to publish, or preparation of the manuscript. M.J.K. is affiliated to the National Institute for Health Research Health Protection Research Unit (NIHR HPRU) in Gastrointestinal Infections at University of Liverpool in partnership with UK Health Security Agency (UKHSA), in collaboration with University of Warwick. M.J.K. is also affiliated to the National Institute for Health Research Health Protection Research Unit (NIHR HPRU) in Genomics and Enabling Data at University of Warwick in partnership with UK Health Security Agency (UKHSA). The views expressed are those of the author(s)

and not necessarily those of the NHS, the NIHR, the Department of Health and Social Care or UK Health Security Agency. We thank Nick Gent at Public Health England for providing access to the data, and thank Ellen Brooks-Pollock for comments on an early draft.

## Author contributions

T.L., E.M.H., R.N.T., M.J.T., M.J.K., and L.D. were involved in conceptualising the study and developed the specific modelling approach. T.L. wrote the model code and undertook the modelling analysis. T.L., A.H., E.S., M.J.T. and M.J.K. curated the underlying data for the model. T.L. and A.H. produced the visualisations for the paper. T.L. wrote the manuscript with input from all other authors. All authors reviewed and edited the manuscript, and all approved the final version.

## Competing interests

We declare no competing interests.
