## [Peer Review File · Nature Communications]

Reviewers' Comments:

Reviewer #1:

Remarks to the Author:

In "Quantifying within-school SARS-CoV-2 transmission and the impact of lateral flow testing in secondary schools in England", the authors present findings from an individually-based transmission model fit to various administrative and testing data to estimate the risk of within-school transmission of SARS-CoV-2 and the possible effectiveness of various testing and isolation strategies. The paper is written very clearly and, for the most part, is accessible to non-modelers. The research is reasonably novel (an extension of a prior model the authors developed to investigate transmission among schoolchildren) and timely.

As discussed below, there are various points where the authors could expand on their assumptions and the related uncertainties in their estimates. I am particularly concerned about the precision with which the authors have estimated the risk of transmission within schools given (1) their model does not necessarily consider within-school risk, but within-age-cohort risk and (2) their model is highly specified but fit to relatively limited data. Given the charged political climate around school openings and transmission risk, these results should be presented with the appropriate context, uncertainty, and assumptions.

(1a) As stated above, it does not appear that this model really measures within school risk vs the risk of schoolchildren to one another, regardless of the location of transmission. Statements such as "We infer that the majority of pupil infections during term time and on school days occurred within school, as opposed to extraneously from the community" do not seem supported by the model structure, which would consider any transmission from, e.g., an 11 year old to 11 year old attending the same school to be 'within school' transmission. This is an important distinction to make when considering counterfactual scenarios; would a reader infer that 49% of cases would have been avoided had schools been closed with no other behavioral changes?

Relatedly, there is no consideration of teacher-pupil within-school transmission, other than the specification that α_1 and α_2 can "account for the impact of indirect infection via teachers and siblings". Are teachers included in the school population, though? How is this facet of within-school transmission considered when calculating the percent of within-school vs external transmission?

(1b) Could the authors explain more of the intuition behind the results for the proportion of transmission occurring within school vs externally? Intuitively, it would seem that students would have had much more limited external contact outside of school in September-October than spring 2021; if we assume similar risk of household infection for both time periods, why then would community risk appear so much lower (relative to school risk) in the fall? I have two concerns: first, that the somewhat arbitrary, constant decrease in compliance in the spring is driving a lot of the increase in within-school transmission proportion in the spring; and second, that perhaps there is an implicit favoring of within school transmission, compared to external transmission. At each time step, are pupils first infected in school, and then if still uninfected able to be infected externally? If risk of both is high (likelier in spring with B.1.1.7), would it be possible that community transmission is being relegated to within-school transmission?

(2) As with many individual based models, this model is highly specified. Twelve parameters are estimated (which parameters are estimated should be stated within the main text). The identifiability of these parameters (and the estimation of these parameters with appropriate uncertainty) is unclear, given the data. E.g., estimating the proportion asymptomatic and their relative transmissibility without information on symptom status. The estimated LFT specificity (>99.9%) seems abnormally high, nor am I sure what signal there is in their data to estimate this value versus the other multipliers on participation rates/testing volume and transmission.

(3) Relatedly, several parameter choices seem likely to underestimate the variability in model outcomes. By definition, $\alpha_0=1$, which appears to indicate certain daily contact between close contacts (i.e. no weekend effects)? Under this assumption, it appears that the equation for $\beta_j(D)$ does not show the probability of transmission, per se, but the probability of contact with

an infected individual. What parameter (maybe K_s) modulates the probability of infection given contact? The close contact size is also fixed within a school; this seems likely to artificially decrease uncertainty/variability in simulations and lead to overly precise estimates of program impact/transmission risk within school.

I also wonder whether the close contact sizes are underestimated in some cases, specifically in schools which did not report a case (in which the contact size was "sampled from the distribution of reported close contact sizes (as a proportion of the school population)". Probabilistically, I would assume smaller schools to be less likely to have a single reported case. I'm not sure that it's appropriate to scale the contact size by school size in this context (bubbling, distant classrooms). Did the authors investigate whether there was an association between school sizes and contact sizes among those schools with a reported case? Are schools without a reported case similar in their distribution of population sizes? How many schools did not report a single case?

(4) The testing/detection process within the model is unclear, particularly in the main text. In model fitting, does $Y_P(k)$ refer to the proportion of schools with a true peak of infected cases, or does it incorporate the testing/detection process described in S4.3? At what rate do students seek PCR testing outside of the mass testing program? Do all symptomatic students immediately seek testing upon symptom onset (I believe so, given S4.1)? This seems optimistic, to say the least. Does the absence of mass testing in Scenario 2 mean that cases could only be detected by at will PCR testing? Are asymptomatic students only detectable through LFT mass testing?

Somewhat related, but it would be helpful especially for non UK-readers to clarify in the main text, not just the supplement, that these are community testing data where 11-16y olds represent school children, not a school-based testing program.

(5) "Our study also assumes uniform transmission rates from infected pupils within any given secondary school" - as I understand it, the primary extension of this model from the authors' previous work is the incorporation of contact structure (separating close contacts, same-year contacts, other-year contacts). Is this true? That would appear to conflict with this statement.

(6) Did the authors investigate what (if any) levels of participation would be necessary for transmission control in Scenario 3 (mass testing only)?

(7) Some minor suggestions for clarification:

- Per public health standards, "isolation" is restricted for suspected or confirmed cases; students asked to remain absent from school due to contact with a suspected or confirmed case, but without being a suspected/confirmed case themselves, would be "quarantined". Is there a reason the authors use "isolation" to refer to what's classically "quarantine"?
- In 279 - is it only relaxation of "distancing" that's of concern, or relaxation of any control measures?
- In Figure S4, is the plot labeled 'c' meant to be labeled 'f'?
- Is it possible to show the model fit to absence data at a more granular scale? e.g., divided by region as in Figure S7?

Reviewer #2:

Remarks to the Author:

In this manuscript, the authors developed a stochastic individual-based model to quantify within-school SARS-CoV-2 transmission and the impact of implemented control measures. They found that twice weekly mass testing using LFTs has helped to control within-school transmission. They also compared the containment strategies and found that repeat testing of close contact alongside mass testing can reduce absences with a marginal increase in within-school transmission. The paper is well-written. Here are my comments and hope these can help improve the current manuscript.

1. UK launched the vaccine roll-out since the beginning of 2021. Is the effect of vaccine considered in this manuscript? Does the vaccine campaign diminish the effectiveness of mass testing using

LFTs on the within-school transmission? For example, if all the parents are vaccinated, the risk of infection for children seem to be low.

2. The authors focus on the 11-16 years old age group. Children aged 12 years and above may be offered the Pfizer/BioNTech vaccine if they are at high risk according to WHO. What is the proportion of vaccinated children in the current study?

3. In this manuscript, the author considered the Alpha variant and the Delta variant was discussed at the end of this paper. It will be meaningful if the author can further study the Delta variant. For example, if Delta presents, dose the repeat testing of close contact alongside mass testing still work? Are there any changes in the parameters? Such as the testing frequency, the percentage of secondary school pupils taking an LFT test.

4. Line 148. The results about the distribution of peak case number across school seems not very surprising. Because the information about this distribution is in the likelihood function.

5. Please add the definition of close contact.

6. Will testing sensitivity change because of the different variants?

Response to reviewers: Quantifying within-school SARS-CoV-2 transmission and the impact of lateral flow testing in secondary schools in England

We thank both reviewers for their thorough and valuable comments. As you will see, we have made substantial changes, both to the main text and through additional modelling, to address the points raised. In particular, we have reframed the paper in terms of 'pupil-to-pupil transmission' rather than 'within-school transmission', and the interpretation of our results has been amended in light of this change. Further, we have reduced the number of parameters inferred in our model from 12 to 10 and have explored the identifiability of remaining parameters by generating synthetic data sets from known parameters, confirming those parameters are recovered from model fitting. We have also improved upon the fitting procedure used for the model - now using an ABC-SMC algorithm, we have increased the number of particles considered at each generation, and we have used wider priors to more fully explore parameter space. We believe we have fully addressed the comments of each reviewer, which we expand upon in turn below.

Reviewers' comments are in blue text while our responses are in black text. Selected sections of added text are in black italicised text, and the line numbers where they appear in the tracked changes manuscript are included. For longer sections of added or amended text, we include the line numbers where they appear in the tracked changes document but are not included in this document.

N.B. The formatting of Table S1 in the tracked changes version has been altered by running latexdiff. For the correctly formatted version of this table S1, see the revised version of the supplement without tracked changes.

Reviewer 1

In "Quantifying within-school SARS-CoV-2 transmission and the impact of lateral flow testing in secondary schools in England", the authors present findings from an individually-based transmission model fit to various administrative and testing data to estimate the risk of within-school transmission of SARS-CoV-2 and the possible effectiveness of various testing and isolation strategies. The paper is written very clearly and, for the most part, is accessible to non-modelers. The research is reasonably novel (an extension of a prior model the authors developed to investigate transmission among schoolchildren) and timely.

As discussed below, there are various points where the authors could expand on their assumptions and the related uncertainties in their estimates. I am particularly concerned about the precision with which the authors have estimated the risk of transmission within schools given (1) their model does not necessarily consider within-school risk, but within-age-cohort risk and (2) their model is highly specified but fit to relatively limited data. Given the charged political climate around school openings and transmission risk, these results should be presented with the appropriate context, uncertainty, and assumptions

1ai) **Comment:** As stated above, it does not appear that this model really measures within school risk vs the risk of schoolchildren to one another, regardless of the location of transmission. Statements such as "We infer that the majority of pupil infections during term time and on school days occurred within school, as opposed to extraneously from the community" do not seem supported by the model structure, which would consider any transmission from, e.g., an 11 year old to 11 year old attending the same school to be 'within school' transmission. This is an important distinction to make when considering

counterfactual scenarios; would a reader infer that 49% of cases would have been avoided had schools been closed with no other behavioral changes?

Response: We thank the reviewer for highlighting this. We agree that our study considers transmission between secondary school pupils rather than transmission within secondary schools in general. Accordingly, we have reframed the manuscript to refer to 'pupil-to-pupil transmission' rather than 'within-school transmission'. Our results regarding within-school vs external transmission have thus been reinterpreted: e.g. *"We infer that the majority of pupil infections during term time and on school days occurred through contact with another pupil."* We also reiterate this point in the discussion:

Added text (lines 358-363 of the main text): *"Results should be interpreted in reference to the underlying model structure. Our model focuses on transmission that occurs between secondary school aged pupils. While one might presume that infection occurs within the secondary school setting, this may not be the case - secondary school pupils will also mix outside of school. This point is particularly relevant in the context of closing schools, which would not be expected to reduce pupil-to-pupil transmission to zero, and may increase mixing between pupils in other settings."*

1aii) **Comment:** Relatedly, there is no consideration of teacher-pupil within-school transmission, other than the specification that α_1 and α_2 can "account for the impact of indirect infection via teachers and siblings". Are teachers included in the school population, though? How is this facet of within-school transmission considered when calculating the percent of within-school vs external transmission?

Response: Teachers are not included explicitly in our model, which is a limitation of our approach. However, we did not feel we had access to sufficient data to parameterise the interaction between pupils and teachers explicitly. The omission of teachers from the model is now stated explicitly in the methods, and is discussed as a limitation in the discussion:

Added text (lines 363-370 of the main text): *"Teachers and other staff members are not included explicitly in our model. Accordingly, our model assumes the impact of transmission from teachers to pupils comprises some proportion of 'external' infections. Previous studies have suggested there is no evidence that teachers are at greater risk of infection or hospitalisation than other key workers (Office for National Statistics 2020, Fenton et al. 2021), though their relative risk may be context dependent. Provided appropriate data were available to parameterise the interaction between teachers and pupils, further research exploring the impact of within-school epidemics on infections in teachers and staff, and conversely the impact of teachers and staff on within-school epidemics, would be valuable."*

1b) **Comment:** Could the authors explain more of the intuition behind the results for the proportion of transmission occurring within school vs externally? Intuitively, it would seem that students would have had much more limited external contact outside of school in September-October than spring 2021; if we assume similar risk of household infection for both time periods, why then would community risk appear so much lower (relative to school risk) in the fall? I have two concerns: first, that the somewhat arbitrary, constant decrease in compliance in the spring is driving a lot of the increase in within-school transmission proportion in the spring; and second, that perhaps there is an implicit favoring of within school transmission, compared to external transmission. At each time step, are pupils first infected in school, and then if still uninfected able to be infected externally? If risk of both is high (likelier in spring with B.1.1.7), would it be possible that community transmission is being relegated to within-school transmission?

Response: In England, stricter population-level control measures were implemented from November 2020 to May 2021 than measures implemented from September-October 2020, with England's second and third national lockdowns commencing on the 5th November 2020 and 6th January 2021 respectively. However, schools remained open to all pupils throughout November and December 2020, and reopened to all pupils from 8th March 2021. Over these periods, positive testing rates within secondary school aged children increased while remaining relatively flat in other age groups. As a consequence, the proportion of positive tests recorded in secondary school aged children out of all positive tests recorded increased. While 5.1% of positive PCR tests were among 11-16 year olds from 31st August 2020 to 31st October 2020, 8.6% of positive PCR tests were in this age groups from 1st November 2020 to 19th December 2021, increasing to 9.1% of positive PCR tests from 8th March 2021 to 21st March 2021. To explain this, our model infers that a higher proportion of transmissions occur between pupils.

Added text (lines 262-276 of the main text): *“Our results demonstrate that many cases in secondary-school aged children likely result from transmission from other secondary school pupils, with such infections comprising approximately 44% of new infections in secondary-school aged children in the September-October 2020 half-term, 71% in the November-December 2020 half-term, and 74% from 8th March to 21st May 2021. These results mirror the trends in community swab testing data collected from the wider population over the corresponding time periods, with a yet higher proportion of total positive PCR tests coming from secondary-school aged children in the three time periods (from Pillar 2 testing data, considering PCR tests excluding confirmatory tests, we calculate that 5.1% of positive PCR tests were from 11-16 year olds from 31st August 2020 to 31st October 2020, 8.6% of positive PCR tests were from 11-16 year olds from 1st November 2020 to 19th December 2021, and 9.1% of positive PCR tests were from 11-16 year olds from 8th March 2021 to 21st March 2021). This trend may be a consequence of the strict population control measures implemented from November-December 2020 and March-May 2021, coupled with schools remaining open over these periods.”*

Further response: Regarding the first specific point raised - an increase in transmission after the October half-term is required to fit PCR testing data in November and December 2020. Prior to the introduction of this parameter, model fitting forced the impact of the new variant to have an extremely large impact (> 100% higher transmissibility). As such a result goes against the consensus surrounding the impact of the B.1.1.7 variant (Davies et al. 2021, Volz et al 2021., Graham et al. 2021), we felt the introduction of this parameter was justified. Even without this parameter, we would arrive at a model with a high R_{school} by the end of the study period. This aspect is now discussed in the supplementary information:

Added text (lines 174-177 of the supplement): *“Without the introduction of this parameter, an increase in transmissibility was attributed to the Alpha variant that we deemed as implausible based on estimates from other studies (Davies et al. 2021, Volz et al 2021., Graham et al. 2021). Because of these correspondence issues, we introduced f to obtain a reasonable fit to Pillar 2 testing data through time, whilst still obtaining increases in transmissibility of the B.1.1.7 variant consistent with contemporary estimates.”*

Further response: Regarding the second specific point raised - each time step, infections from other pupils and infections occurring from community transmission happen independently from one another. However, we did previously record those infected by both sources as occurring via within-school

transmission. The proportion of infections that occur from both sources are very small (<0.03%). Nevertheless, this has been adjusted such that infections occurring from both sources are assigned randomly as either occurring from pupil-to-pupil transmission or occurring from an external source. This correction does not qualitatively impact our results. This point has been clarified in the supplementary methods:

Added text (lines 107-111 of the supplement): *“External transmission to pupils and pupil-to-pupil transmission occur independently each time step. If a pupil is infected both via external transmission and from another pupil on the same day, the pupil is assigned randomly as internally or externally infected. Pupils infected via both routes occur only rarely in simulations, accounting for less than 0.03% of infections in the simulations from the main analysis.*

2) **Comment:** As with many individual based models, this model is highly specified. Twelve parameters are estimated (which parameters are estimated should be stated within the main text). The identifiability of these parameters (and the estimation of these parameters with appropriate uncertainty) is unclear, given the data. E.g., estimating the proportion asymptomatic and their relative transmissibility without information on symptom status. The estimated LFT specificity (>99.9%) seems abnormally high, nor am I sure what signal there is in their data to estimate this value versus the other multipliers on participation rates/testing volume and transmission.

Response: We agree with the reviewer that the identifiability of parameters is an important issue for any individual-based model, and that minimising the number of parameters inferred should be a priority. To address this, we have reduced the number of parameters estimated from twelve to ten (which are now stated within the main text). The external transmission scaling constant is now no longer drawn from a lognormal distribution, and hence σ_{eps} is no longer required, while we also fix LFT specificity. We note that the reviewer states that the estimated LFT specificity is abnormally high. However, LFT specificity from school testing in England has been extremely high - with reports stating an LFT specificity of at least 99.9% (Funk et al. 2021, DHSC and Public Health England 2021). We now use the specificity estimated by the Department for Health and Social Care and Public Health England (2021) of 99.97% in our model. Additionally, we widened the priors for the parameters K , ϵ , and f to more fully explore the possible combinations of relative levels of external vs pupil-to-pupil transmission. To assess the identifiability of the remaining model parameters, we fit our model, now using a ABC-SMC method, to five synthetic data sets generated from known parameters. All parameters were recovered during this exercise. The posterior density plots of these results, and a discussion of identifiability of parameters is included in Supplement S5.4. Where relevant, we have highlighted the interpretation of our model parameters as conditional on modelling assumptions (e.g. ***“With an assumed LFT specificity of 99.97%, the data were best explained by a model that assumed only 38% of negative home LFTs are reported.”***). We have included in our discussion caveats regarding inferred parameters:

Added text (lines 371-380 of the main text) : *“Similarly, inferred parameters should be interpreted cautiously and in the context of the underlying model assumptions. For example, the model fits the proportion of infected secondary school pupils who are symptomatic and the relative infectiousness of asymptomatic individuals, under the assumption that all symptomatic individuals seek a test upon symptom onset. In reality, not all symptomatic individuals will seek a test, and may remain in school throughout their infectious period. Consequently, the ‘true’ proportion of pupils who are symptomatic will be higher than the value obtained through model fitting, while the ‘true’ relative infectiousness of*

asymptomatic individuals will be lower. Similarly, levels of underreporting are inferred under the assumption that LFTs during mass testing schools have had a 99.97% specificity (in line with contemporary estimates (Funk et al. 2021, DHSC and Public Health England 2021)). Lower assumed LFT specificity would yield lower levels of underreporting, as more false positives would occur for a lower number of tests.”

3j) **Comment:** Relatedly, several parameter choices seem likely to underestimate the variability in model outcomes. By definition, $\alpha_0=1$, which appears to indicate certain daily contact between close contacts (i.e. no weekend effects)? Under this assumption, it appears that the equation for $\beta_j(D)$ does not show the probability of transmission, per se, but the probability of contact with an infected individual. What parameter (maybe K_s) modulates the probability of infection given contact? The close contact size is also fixed within a school; this seems likely to artificially decrease uncertainty/variability in simulations and lead to overly precise estimates of program impact/transmission risk within school.

Response: We thank the reviewer for highlighting these items requiring clarification. The text and equations in Supplementary Text S3 has been updated and amended for clarification (lines 86-214 of the supplement). The subsection on pupil-to-pupil transmission (previously within-school transmission) now comes before school contact structure. $I_i(d)$ has been reassigned as $\lambda_i(d)$, and its explicit interpretation has been given: **“the expected number of secondary infections from individual i on day d, assuming all their contacts are susceptible”**. Weekend effects are included in the model, and are now indicated in the equation for $\lambda_i(d)$ (previously $I_i(d)$). We also include an equation for $\tau(i,j)$, which details the probability that an infectious individual i infects a susceptible individual j on day d, with $\tau(i,j) = 0$ if j is not attending school stated explicitly. K_s modulates the probability of infection given a contact, but is a composite term that captures both the probability of infection given a contact and an individual’s contact rate. Alpha terms refer to the relative probability of contact with individuals, e.g. the probability of contact with a given non-close contact member in your year group is α_1 times the probability of contact with a given close contact in your year group.

Further response: We agree that fixing close contact group sizes limits the heterogeneity in contact structures within schools. However, we did not believe there was sufficient data to properly parameterise the contact structure of different schools. Fixing close contact sizes for each school was assumed on the basis that a) the context of different schools likely means the number of close contacts isolated given a confirmed case varies between schools, and that b) capturing this is important to capture the proportion of pupils that are absent through time. While the assumption we make is a simplifying one, we believe that our model successfully captures trends in school absences through time, despite the simplicity of the approach. The limitations of this assumption, the decrease in variability that results from this assumption, and a discussion of previous studies recording the social structure of contacts within schools, are now included in the discussion.

Added text (lines 316-318 of the main text): *“Assuming that schools implemented consistent isolation policies throughout each term, we also assumed that close contact group sizes were of a fixed size for each school. While this assumption allowed us to successfully match to absence data throughout both terms, we acknowledge it limits the heterogeneity in contact patterns within schools.”*

Further added text (lines 322-328 of the main text): *“While previous studies undertaken prior to the COVID-19 pandemic have attempted to record contact mixing patterns within schools (Salathe et al. 2010, Eames et al. 2011, Conlan et al. 2011), the implementation of rigid social distancing measures*

within schools mean that such studies are not of direct use in the context of COVID-19. The CoMix study has surveyed social contacts in the UK during the COVID-19 pandemic and has been used to infer age-dependent mixing matrices (Jarvis et al. 2020), though these data are not directly informative of contact structure within schools. A deeper understanding of the interplay between contact network structure within schools and the success of control measures would be an important contribution going forward.”

3ij) **Comment:** I also wonder whether the close contact sizes are underestimated in some cases, specifically in schools which did not report a case (in which the contact size was "sampled from the distribution of reported close contact sizes (as a proportion of the school population)". Probabilistically, I would assume smaller schools to be less likely to have a single reported case. I'm not sure that it's appropriate to scale the contact size by school size in this context (bubbling, distant classrooms). Did the authors investigate whether there was an association between school sizes and contact sizes among those schools with a reported case? Are schools without a reported case similar in their distribution of population sizes? How many schools did not report a single case?

Response: We thank the reviewer for raising this point. Different schools appear to have taken different approaches after a confirmed case - some appear to have isolated only close contacts (which as the reviewer points out, does not scale with school size), while others appear to have isolated year groups or some fraction of the school (which does scale with school size). The relationship between the close contact sizes inferred from absences data and school size is now captured in Supplementary figure S1. Supplementary figure S1 shows that for the majority of schools, close contact group size does not scale with school size. Accordingly, close contact group sizes for schools which did not report a case are now sampled from the distribution of reported close contact sizes, as numbers rather than as a proportion. This has not qualitatively impacted our results. The proportion of schools that had reported a single case, and the difference in distributions between those reporting one case and those who have not reported one case are now included in Supplementary text S2 (lines 70-78 of supplement).

4) **Comment:** The testing/detection process within the model is unclear, particularly in the main text. In model fitting, does $Y_P(k)$ refer to the proportion of schools with a true peak of infected cases, or does it incorporate the testing/detection process described in S4.3? At what rate do students seek PCR testing outside of the mass testing program? Do all symptomatic students immediately seek testing upon symptom onset (I believe so, given S4.1)? This seems optimistic, to say the least. Does the absence of mass testing in Scenario 2 mean that cases could only be detected by at will PCR testing? Are asymptomatic students only detectable through LFT mass testing?

Somewhat related, but it would be helpful especially for non UK-readers to clarify in the main text, not just the supplement, that these are community testing data where 11-16y olds represent school children, not a school-based testing program.

Response: A summary of the testing process has now been added to the main text (lines 104-122 of the main text), and we have clarified several aspects of the testing process in Supplementary text S4. $S_P(k)$ refers to the proportion of schools with a peak of k confirmed cases over time period P , which has now been stated explicitly in Supplementary text S5. Asymptomatic students can only be detected by mass testing, and all symptomatic students immediately sought a test upon symptom onset, which are now both stated explicitly in Supplementary text S4. The limitations and implications of assuming all

symptomatic students seek a test upon symptom onset are now commented upon in the discussion. We have also clarified in the main text that testing data are from community testing data in 11-16 year olds:

Added text (lines 123-125 of the main text): *"The model was fitted to swab testing data recorded in 11-16 year olds collected in the wider population from both PCR tests and LFTs, referred to as Pillar 2 data, which we took as a proxy for positive testing rates within secondary school pupils."*

5) **Comment:** "Our study also assumes uniform transmission rates from infected pupils within any given secondary school" - as I understand it, the primary extension of this model from the authors' previous work is the incorporation of contact structure (separating close contacts, same-year contacts, other-year contacts). Is this true? That would appear to conflict with this statement.

Response: We agree that the above statement is unclear, and apologise for any confusion. This statement referred to the lack of variation in infectiousness of individuals within a school, but you are correct, within a school a pupil transmits infection to their close contacts, other year group contacts, and other year contacts at different rates. This has now been clarified as:

Added text (lines 314-319 of the main text): *"Our study assumed three levels of mixing, with pupils transmitting infection at high rates to their close contacts, at a lower rate to other pupils in their year, and at a yet lower rate to pupils in other years ... In reality, transmission rates are likely to be heterogeneous within schools, both as a consequence of heterogeneous contact patterns and because transmission rates are likely a function of peak viral load (He et al. 2020), which varies between individuals."*

6) **Comment:** *Did the authors investigate what (if any) levels of participation would be necessary for transmission control in Scenario 3 (mass testing only)?*

Response: Thanks for this suggestion. This has now been added into Figure 4D and is stated in the results.

7) **Comment:** Some minor suggestions for clarification:

7i) **Comment:** Per public health standards, "isolation" is restricted for suspected or confirmed cases; students asked to remain absent from school due to contact with a suspected or confirmed case, but without being a suspected/confirmed case themselves, would be "quarantined". Is there a reason the authors use "isolation" to refer to what's classically "quarantine"?

Response: Throughout the course of the COVID-19 pandemic, the UK government has referred to the quarantine of pupils who may have been close contacts of other pupils as 'isolation' or 'self-isolation'. A brief justification of the choice of terminology, and an acknowledgment of the alternative terminology is now included in Supplementary text S4:

Added text (lines 263-266 of the supplement): *"Upon a pupil testing positive to a PCR test, the close contacts of that pupil did not attend school for ten days counted from the day of last contact. We refer to this as the isolation of close contacts, consistent with the language used to describe school policies in the event of a confirmed case in the UK, though this could also be described as the quarantine of close contacts."*

7ii) **Comment:** In 279 - is it only relaxation of "distancing" that's of concern, or relaxation of any control measures?

Response: Thanks for highlighting this - we have now changed the text to refer to control measures rather than distancing measures.

7iii) **Comment:** In Figure S4, is the plot labeled 'c' meant to be labeled 'f'?

Response: This has now been amended.

7iv) **Comment:** Is it possible to show the model fit to absence data at a more granular scale? e.g., divided by region as in Figure S7?

Response: We now include the model comparison to absence data at a regional level in Supplementary Figure S18, and briefly discuss regions and periods where there are discrepancies between the models.

Reviewer 2

In this manuscript, the authors developed a stochastic individual-based model to quantify within-school SARS-CoV-2 transmission and the impact of implemented control measures. They found that twice weekly mass testing using LFTs has helped to control within-school transmission. They also compared the containment strategies and found that repeat testing of close contact alongside mass testing can reduce absences with a marginal increase in within-school transmission. The paper is well-written. Here are my comments and hope these can help improve the current manuscript.

1) **Comment:** UK launched the vaccine roll-out since the beginning of 2021. Is the effect of vaccine considered in this manuscript? Does the vaccine campaign diminish the effectiveness of mass testing using LFTs on the within-school transmission? For example, if all the parents are vaccinated, the risk of infection for children seem to be low.

Response: While the effects of the adult population being vaccinated is not explicitly included in the model, the impact of the vaccine roll-out in adults is indirectly captured by the structure of the model, as vaccine uptake in a local area will impact community prevalence, which determines the probability of external transmission to pupil's through time. However, the model does not account for heterogeneities in vaccine uptake within local communities which may impact a pupil's probability of external infection. This is now included in the discussion:

Added text (lines 353-357 of the main text): *"The impact of vaccination on external transmission is implicit in our model, as the probability of external transmission depends upon community infections, which in turn depends upon local vaccine uptake. However, we do not account for heterogeneities in vaccine uptake within an LTLA, which may impact a pupil's probability of external infection."*

2) **Comment:** The authors focus on the 11-16 years old age group. Children aged 12 years and above may be offered the Pfizer/BioNTech vaccine if they are at high risk according to WHO. What is the proportion of vaccinated children in the current study?

Response: In the model, we assume that no children are vaccinated. We believe this to be a reasonable simplifying assumption, as prior to 21st May 2021, only 4.2% of 16-17 year olds, and 0.02% of 12-15 year olds had received a vaccine. However, as the vaccination has since been approved for all 12-16 year olds, the inclusion of vaccination explicitly may be an important factor in any future modelling work, which is now elaborated on in the discussion.

Added text (lines 344-353 of the main text): *“By 21st May 2021, no COVID-19 vaccine had been approved in the UK for general use for pupils aged 12-17 years, although vaccines were available to children within that age group who were classed as clinically vulnerable and those aged 18 years. In England, 0.02% of 12-15 year olds and 4.2% of 16-17 year olds had received at least one dose of vaccine by 21st May 2021 (UK Government, 2021). A higher proportion of 18 year olds had likely received one dose by this date, as 16.7% of 18-24 year olds had received at least one dose, though not all 18 year olds attend secondary school. Because vaccine uptake prior to 21st May 2021 was relatively limited amongst secondary school pupils, we would not expect the action of vaccination to have had a substantive impact on our results. Since 13th September 2021, a vaccine has been approved for all 12-17 year olds and, consequently, the inclusion of vaccination explicitly may be an important factor in future models of transmission within secondary schools.”*

3) **Comment:** In this manuscript, the author considered the Alpha variant and the Delta variant was discussed at the end of this paper. It will be meaningful if the author can further study the Delta variant. For example, if Delta presents, does the repeat testing of close contact alongside mass testing still work? Are there any changes in the parameters? Such as the testing frequency, the percentage of secondary school pupils taking an LFT test.

Response: While we believe the inclusion of the Delta variant is beyond the remit of this study, we have expanded upon the questions asked in the discussion:

Added text (lines 380-385 of the main text): *“Model parameters may be impacted by the emergence of new variants not included in the model, such as the Delta (B.1.617.2) variant. More transmissible variants would be expected to increase pupil-to-pupil transmission rates, but may also impact the relative infectiousness of asymptomatic pupils and the proportion of pupils who display symptoms if the new variant changes the symptomatology of the virus (though no significant changes in symptomatology were observed between the wild-type and the Alpha variant (Graham et al. 2021))”*

4) **Comment:** Line 148. The results about the distribution of peak case number across school seems not very surprising. Because the information about this distribution is in the likelihood function.

Response: We now clarify that this is something explicitly fitted to - added text in bold:

Added text (in bold) (lines 184-187 of the main text): *“We note that the model slightly underestimates the proportion of schools that had a low peak number of confirmed cases during the September-December 2020 period, while overestimating the proportion of schools with a low peak number of confirmed cases from March to May 2021, **despite being explicitly fitted to these data sources.**”*

5) **Comment:** Please add the definition of close contact.

Response: The definition of a close contact has now been included:

Added text (lines 114-119 of the main text): *“Close contact group sizes were defined as the number of pupils asked to self-isolate upon identification of a positive case. Schools were assumed to implement consistent isolation policies throughout each term, meaning that close contact group sizes were fixed within each school. Close contact group sizes were inferred from Department for Education: Education Setting Status data in order to capture the proportion of absences that occur through time (Supplementary Text S2).”*

6) **Comment:** *Will testing sensitivity change because of the different variants?*

Response: Different variants may impact test sensitivity. The following text has been added to the discussion:

Added text (lines 385-386 of the main text): *“Relatedly, test sensitivity and specificity may be impacted by the emergence of new variants (Del et al. 2021).”*

Reviewers' Comments:

Reviewer #1:

Remarks to the Author:

I thank the authors for their extensive revisions and detailed response to my previous comments. I have just one remaining minor comment, regarding the use of weekend effects (the conditions of $\lambda(d)$).

If I understand correctly, weekend (or non school-term) transmission between pupils occurs only as external infections -- if so, it would be worthwhile to explicitly state this assumption (e.g., "Our model focuses on transmission that occurs between secondary school aged pupils on school days"). I don't wish to belabor the point unnecessarily, but hope to be conscious of the wider implications and interpretations of this work. Thanks again to the authors for their gracious responses.

Reviewer #2:

Remarks to the Author:

In this version of the manuscript, the authors addressed all my questions except one about the Delta.

To compare four different strategies for controlling pupil-to-pupil transmission, the authors performed simulation over the period 1st March 2021 (the week prior to schools reopening after the 2021 lockdown) to 23rd May 2021 (when the Delta variant began to spread widely). Based on these simulations, the authors concluded that a strategy of repeat testing of close contacts rather than isolation, alongside mass testing, substantially reduces absences with only a marginal increase in within-school pupil-to-pupil transmission.

This conclusion holds for Alpha. However, after 23rd May 2021, the Delta variant began to spread widely. Does this conclusion still hold for Delta? The Delta analysis will gain some insights for control measures specific to pupils in practices. It is better to keep the analysis up to date, given that Alpha has been replaced by Delta.

Response to Reviewers: Quantifying pupil-to-pupil SARS-CoV-2 transmission and the impact of lateral flow testing in secondary schools in England

Reviewer 1:

Comment: I thank the authors for their extensive revisions and detailed response to my previous comments. I have just one remaining minor comment, regarding the use of weekend effects (the conditions of $\lambda(d)$). If I understand correctly, weekend (or non school-term) transmission between pupils occurs only as external infections – if so, it would be worthwhile to explicitly state this assumption (e.g., "Our model focuses on transmission that occurs between secondary school aged pupils on school days"). I don't wish to belabor the point unnecessarily, but hope to be conscious of the wider implications and interpretations of this work. Thanks again to the authors for their gracious responses.

Response: We have now explicitly stated that our model focuses on transmission between secondary school aged pupils on school days both in the main text (Method section) , and in the supplementary information (Supplementary Text 3).

Added text (Main text, Methods section): *"Our model focused on transmission between secondary school pupils on school days"*

Added text (Supplementary Text 3): *"On weekends or on days when schools are not open, transmission to pupils occurs only via external infection."*

Reviewer 2:

Comment: In this version of the manuscript, the authors addressed all my questions except one about the Delta. To compare four different strategies for controlling pupil-to-pupil transmission, the authors performed simulation over the period 1st March 2021 (the week prior to schools reopening after the 2021 lockdown) to 23rd May 2021 (when the Delta variant began to spread widely). Based on these simulations, the authors concluded that a strategy of repeat testing of close contacts rather than isolation, alongside mass testing, substantially reduces absences with only a marginal increase in within-school pupil-to-pupil transmission. This conclusion holds for Alpha. However, after 23rd May 2021, the Delta variant began to spread widely. Does this conclusion still hold for Delta? The Delta analysis will gain some insights for control measures specific to pupils in practices. It is better to keep the analysis up to date, given that Alpha has been replaced by Delta.

Response: We fully appreciate the desire to include the latest information in any paper, and future work will infer parameters for Delta, as well as consider the impact of childhood vaccination. However, we strongly feel that focusing on the period where the Alpha variant predominated in schools provides a coherent narrative - including Delta would not only double the amount of parameter inference required, but would also substantially increase the complexity of the paper as each result would need to be

investigated for both the Alpha and Delta variants. Furthermore, additional changes to the model might be required to capture events during the school academic year that can impact attendance, independent of the epidemiological situation (for example, accounting for changes in attendance in summer term during exam season). We therefore believe that an extension to the model accounting for the impact of the Delta variant on within-school transmission is beyond the scope of this paper.

Nonetheless, we agree with the reviewer that it would be interesting to explore the effects of different variants (not only including Delta, but also the recent emergence of the Omicron variant) on our results. We have therefore suggested in our Discussion that further work considering the impact of the Delta wave (and potentially the current Omicron wave) would be a valuable line of future research. We have also discussed the potential impact of Delta on our results in the discussion, including commenting on the potential impacts of features of different variants (the increased transmissibility of variants, altered symptomatology of variants, testivity sensitivity and specificity, and altered generation time distributions of new variants) in the context of our results.

Added text (Main text, Discussion section), added text in bold: *“Model parameters may be impacted by the emergence of new variants not included in the model, such as the Delta (B.1.617.2) variant. More transmissible variants would be expected to increase pupil-to-pupil transmission rates, but may also impact the relative infectiousness of asymptomatic pupils and the proportion of pupils who display symptoms if the new variant changes the symptomatology of the virus (though no significant changes in symptomatology were observed between the wild-type and the Alpha variant (Graham et al. 2021). **Relatedly, the emergence of new variants may impact the generation time distribution of SARS-CoV-2 (Hart et al. 2021), which informs the infectiousness profile of individuals within the model, while test sensitivity and specificity may also be impacted (Del et al. 2021). Such changes may impact the relative effectiveness of different control measures.**”*

Added text (Main text, Discussion section), added text in bold: *“With R_{school} approximately equal to 1 in mid-May 2021, more transmissible variants such as the Delta variant (estimated to be 60% more transmissible than the Alpha variant (Public Health England, 2021) could increase R_{school} , substantially above 1. **If such an increase were to occur, within-school epidemics would occur more frequently. In turn, this may result in high levels of absences as a consequence of high numbers of cases amongst pupils.**”*

Added text (Main text, Discussion section): *“Future studies considering the impact of Delta and future variants on transmission within secondary schools, and whether strategies utilising mass testing remain capable of mitigating transmission while keeping absence levels low, would be a valuable line of further research.”*